# Modulation of large rhythmic depolarizations in human large basket cells by norepinephrine and acetylcholine
Danqing Yang[1,2], Guanxiao Qi [1], Jonas Ort[3,4,5], Victoria Witzig [6], Aniella Bak [7], Daniel Delev[3,4,5], Henner Koch[7] & Dirk Feldmeyer [1,2,8] ✉

Rhythmic brain activity is critical to many brain functions and is sensitive to neuromodulation, but so far very few studies have investigated this activity on the cellular level in vitro in human brain tissue samples. This study reveals and characterizes a novel rhythmic network activity in the human neocortex. Using intracellular patch-clamp recordings of human cortical neurons, we identify large rhythmic depolarizations (LRDs) driven by glutamate release but not by GABA. These LRDs are intricate events made up of multiple depolarizing phases, occurring at ~0.3 Hz, have large amplitudes and long decay times. Unlike human tissue, rat neocortex layers 2/3 exhibit no such activity under identical conditions. LRDs are mainly observed in a subset of L2/3 interneurons that receive substantial excitatory inputs and are likely large basket cells based on their morphology. LRDs are highly sensitive to norepinephrine (NE) and acetylcholine (ACh), two neuromodulators that affect network dynamics. NE increases LRD frequency through β-adrenergic receptor activity while ACh decreases it via $M_4$ muscarinic receptor activation. Multi-electrode array recordings show that NE enhances and synchronizes oscillatory network activity, whereas ACh causes desynchronization. Thus, NE and ACh distinctly modulate LRDs, exerting specific control over human neocortical activity.

Rhythmic brain activity plays a pivotal role in numerous brain functions, from sensory processing to memory consolidation[1–4]. Over the past few decades, several studies have shown that resected human cortical tissue can sustain rhythmic network activities across various frequency bands[5–9]. Apart from oscillations elicited by pharmacological application or electrical stimulation[6,8,10], spontaneous synchronous events were found in slices prepared from human epileptic neocortex and were suspected to be epilepsy-related[5–7,9]. Intriguingly, comparable rhythmic activity has been observed in the healthy monkey hippocampus and non-epileptic neocortex[9,11,12], suggesting that these events may indeed be correlated with physiological network activity independent of epileptogenic processes. It is worth noting that most of these rhythmic brain activity were recorded using an artificial cerebrospinal fluid (ACSF) with a higher concentration of $K^+$ to increase the overall excitability of the brain slices. The use of a similar ACSF in the ferret, mouse, and rat neocortex have been linked to the induction of in vitro slow oscillations, known as 'Up' states, leading to discharges in neuronal populations[13–15]. Consequently, the diverse characteristics of in vitro brain oscillations present challenges in pinpointing the functional mechanisms that drive them.

Although spontaneous synchronous activity has been previously described in human cortical slices, these activities were usually measured using multiple-electrode array or sharp microelectrode techniques, making it hard to attain high resolutions for deciphering their temporal structures and their associations with non-synchronous events[5,6,9,12]. Additionally, the specific morphology of individual neurons participating in these network activities has been largely unexplored. Unraveling whether particular neuronal cell types are disproportionately active in these networks is essential, as they may be instrumental in mediating state transitions in the brain, with potential significant implications for behavior and cognition.

[1]Research Center Juelich, Institute of Neuroscience and Medicine 10, Research Center Juelich, 52425 Juelich, Germany. [2]Department of Psychiatry, Psychotherapy, and Psychosomatics, RWTH Aachen University Hospital, 52074 Aachen, Germany. [3]Department of Neurosurgery, Faculty of Medicine, RWTH Aachen University Hospital, Aachen, Germany. [4]Neurosurgical Artificial Intelligence Laboratory Aachen (NAILA), RWTH Aachen University Hospital, 52074 Aachen, Germany. [5]Center for Integrated Oncology, Universities Aachen, Bonn, Cologne, Düsseldorf (CIO ABCD), Bonn, Germany. [6]Department of Neurology, RWTH Aachen University Hospital, 52074 Aachen, Germany. [7]Department of Neurology, Section Epileptology, RWTH Aachen University Hospital, 52074 Aachen, Germany. [8]Jülich-Aachen Research Alliance, Translational Brain Medicine (JARA Brain), Aachen, Germany. ✉e-mail: d.feldmeyer@fz-juelich.de

Norepinephrine (NE) and acetylcholine (ACh) are neuromodulators historically known to influence network dynamics across varying behavioral states[16–18]. By stimulation of brainstem noradrenergic and cholinergic nuclei or the basal forebrain, NE and ACh release is known to modulate cortical oscillations[19–22]. Yet, our understanding of how NE and ACh modulate the human neocortex remains limited[23,24], because our current knowledge about the neuromodulation of the neocortex by NE and ACh is derived almost exclusively from in vitro/in vivo experiments in rodents. Given that primates, including humans, exhibit a higher density of neuromodulatory afferents in the neocortex than rodents[25,26], it is essential to investigate the neuromodulation of neuronal network activity in human neocortical tissue to elucidate the mechanisms at play. Here, we report that in acute human brain slices, spontaneous large rhythmic depolarizations (LRDs) were detected in neocortex in the absence of pharmacological manipulation or external stimulation. These LRDs are complex network events that depend on presynaptic glutamatergic release but are independent of GABAergic release. LRDs were more frequent in human interneurons than in pyramidal cells; these were not observed in rat brain slices under similar conditions. Further characterization revealed that human interneurons exhibiting LRDs (LRD + ) have distinct, broad dendritic and axonal arborization patterns, and are of the large basket cell type. This is in stark contrast to interneurons that show no LRD activity (LRD-). Moreover, LRDs are highly sensitive to noradrenergic and cholinergic modulation with NE enhancing and ACh reducing LRD frequency. MEA recordings revealed a clear increase in neuronal spiking synchronization by NE while ACh drove neurons towards a more desynchronized firing pattern. This implies that synchronous brain activity is under antagonistic neuromodulatory control of NE and ACh in the human brain.

## Results

### Large rhythmic depolarizations (LRDs) are observed in a subset of human cortical L2/3 neurons

Acute brain slices were prepared from tissue blocks following brain surgery (resection of epileptic and tumor foci) in 24 patients aged 8–75 years. The neocortical tissue used predominantly came from the temporal or frontal cortex, with exceptions including one case from the occipital and another from the parietal cortex (Supplementary Tab. S1). The tissue was removed during surgical access to the pathological brain region and was sufficiently distant from the tumor and/or epileptogenic focus. Whole-cell current clamp recordings were made in human cortical L2/3 neurons with simultaneous biocytin filling, which allowed post hoc identification of their morphology. 82 pyramidal cells (PCs) and 64 interneurons (INs) were identified by their electrophysiological properties and morphological features (Fig. 1e). In a subset of human L2/3 neurons (24 out of 146 neurons), large rhythmic depolarizations (LRDs) were observed against a background of excitatory postsynaptic potentials (EPSPs). In some neurons (9 out of 17 L2/3 LRD+ interneurons), the depolarization triggered direct action potential (AP) discharges (Fig. 1a). LRDs were not associated with a specific brain region, gender, age or pathophysiology (Supplementary Tab. S1). For each neuron exhibiting LRDs, we analyzed the excitatory postsynaptic activity (continuous current-clamp recording, 100 s). LRDs and EPSPs were analyzed separately. In L2/3 interneurons, LRDs exhibited an average amplitude of $10.2 \pm 2.9$ mV ($n = 535$ events in 17 neurons), which was significantly larger than that of unitary EPSPs ($2.0 \pm 1.0$ mV, $n = 17213$ events in 17 neurons). Dramatic differences in dynamic properties were also observed between LRDs and EPSPs. The decay time was significantly longer for LRDs ($152.5 \pm 43.3$ ms) than for EPSPs ($39.9 \pm 13.6$ ms) (Fig. 1b, d). LRDs occurred at a low frequency ranging between 0.08 and 0.70 Hz (mean = $0.29 \pm 0.18$ Hz). In contrast, EPSPs had a much higher frequency of $9.7 \pm 5.7$ Hz (Fig. 1c, d) suggesting that the functional mechanisms underlying LRD generation are fundamentally different from those of EPSPs. LRDs were observed more frequently in L2/3 interneurons than in PCs (26.6% or 17 out of 64 for interneurons vs. 8.5% or 7 out of 82 for PCs; Fig. 1f). LRD frequency and rise time in human L2/3 PCs and interneurons were similar, however, L2/3 PCs exhibited a smaller LRD amplitude and a

prolonged decay time (Supplementary Fig. S1). No spontaneous AP firing was triggered by LRDs in all 7 PCs showing LRDs. We found that LRDs were detected only within the first 6 hours after slicing but not thereafter (see also Supplementary Fig. S2).

To investigate whether LRDs were also present in rodents, recordings were performed from L2/3 neurons in acute brain slices of adult rat prefrontal and temporal cortex. No LRDs activity was detected in L2/3 PCs ($n = 18$, Fig. 1f & Supplementary Fig. S1). We observed sporadic LRD-like events in 2 out of 21 L2/3 interneurons but these events showed no rhythmicity or persistency and therefore were markedly different from typical LRDs observed in human neurons (Supplementary Fig. S1). Our data suggest that the human brain is more likely to generate LRDs than the rat brain under the same recording conditions.

It has been reported that slow rhythmic activity is initiated and prominent in infra-granular layers and then propagates to supra-granular layers[13]. To investigate network activity across all neocortical layers, we performed extracellular recordings on cultured human cortical brain slices using a 256 channel multi-electrode array (MEA). Local field potentials (LFPs) were detected both in L2/3 and L5/6 (Supplementary Fig. S3a, b). Whole-cell recordings were carried out in human L6 cortical neurons; typical LRD activity was also observed in L6 interneurons (Supplementary Fig. S3c, d). Most of the network events detected on the MEA originated in deep layers (L5/6, 73.1%), with the remaining events originating in L2/3 (26.9%) (Supplementary Fig. S3e). The spatial activity pattern of the events propagates in both horizontal (within layers) and vertical directions (across layers) in a complex manner (Supplementary Fig. S3f, g). Taken together, LRDs are not confined to L2/3 but occur in different neocortical layers.

Previous studies suggested that human L2/3 PCs form stronger, more reliable connections compared to rodents[27,28]. Such strong connections make a direct postsynaptic AP more likely and could contribute to synchronized AP firing in the local neuronal microcircuitry. Contrary to spontaneous unitary synaptic inputs triggered by individual presynaptic APs, we hypothesize that LRDs are induced by near-synchronous firing in presynaptic glutamatergic neurons (Fig. 2a). As a first test of this hypothesis, we performed a time-frequency analysis of the spontaneous synaptic activity in LRD+ interneurons. Low frequency LRDs and high frequency EPSPs can be clearly discriminated (Fig. 2b). The onset of a LRD was always accompanied by a superposition of small amplitude events, indicating that LRDs are network-driven synchronous oscillations (Fig. 2b). To reveal the temporal structure of LRDs, we performed voltage clamp recordings in neurons that exhibited LRDs. During the course of an LRD, we observed a predominance of EPSCs with a minimal contribution from IPSCs (Fig. 2c, d).

To investigate the presynaptic mechanisms underlying LRD generation, either 0.5 μM TTX or 10 μM CNQX was bath-applied. Both TTX ($n = 2$) and CNQX ($n = 3$) completely eliminated LRDs, suggesting that LRDs are AP-dependent synaptic events relying on presynaptic glutamate release (Fig. 2e). Previous studies have reported so-called 'giant depolarizing events' for rat hippocampus and neocortex during the first postnatal week, and were elicited by γ-aminobutyric acid (GABA) which is a depolarizing transmitter at early developmental stages because of a high intracellular $Cl^-$ concentration[29–32]. In our experiments bath application of gabazine (1 μM, $n = 4$) did not alter the LRD amplitude, frequency or dynamic properties (Fig. 2e and Supplementary Fig. S4), indicating that GABAergic synaptic transmission do not play a causal role in LRD generation.

Since LRD+ neurons might represent a group of highly connected cells in the neuronal network, we examined the timing and magnitude of EPSPs in L2/3 interneurons that did and did not show LRDs. Notably, LRD+ interneurons exhibited a much higher EPSP frequency compared to LRD- interneurons ($11.0 \pm 7.7$ vs. $2.9 \pm 2.4$ Hz, $P < 0.001$). In addition, the amplitude of spontaneous EPSP was on average 1.5 times larger in LRD+ interneurons compared to LRD- interneurons (LRD + INs: $2.1 \pm 0.9$ mV, LRD - INs: $1.4 \pm 0.5$ mV, $P < 0.05$) (Fig. 2f–h). In summary, our data demonstrates that LRD+ interneurons receive robust excitatory input and participate in neuronal network activity. This supports the notion that the

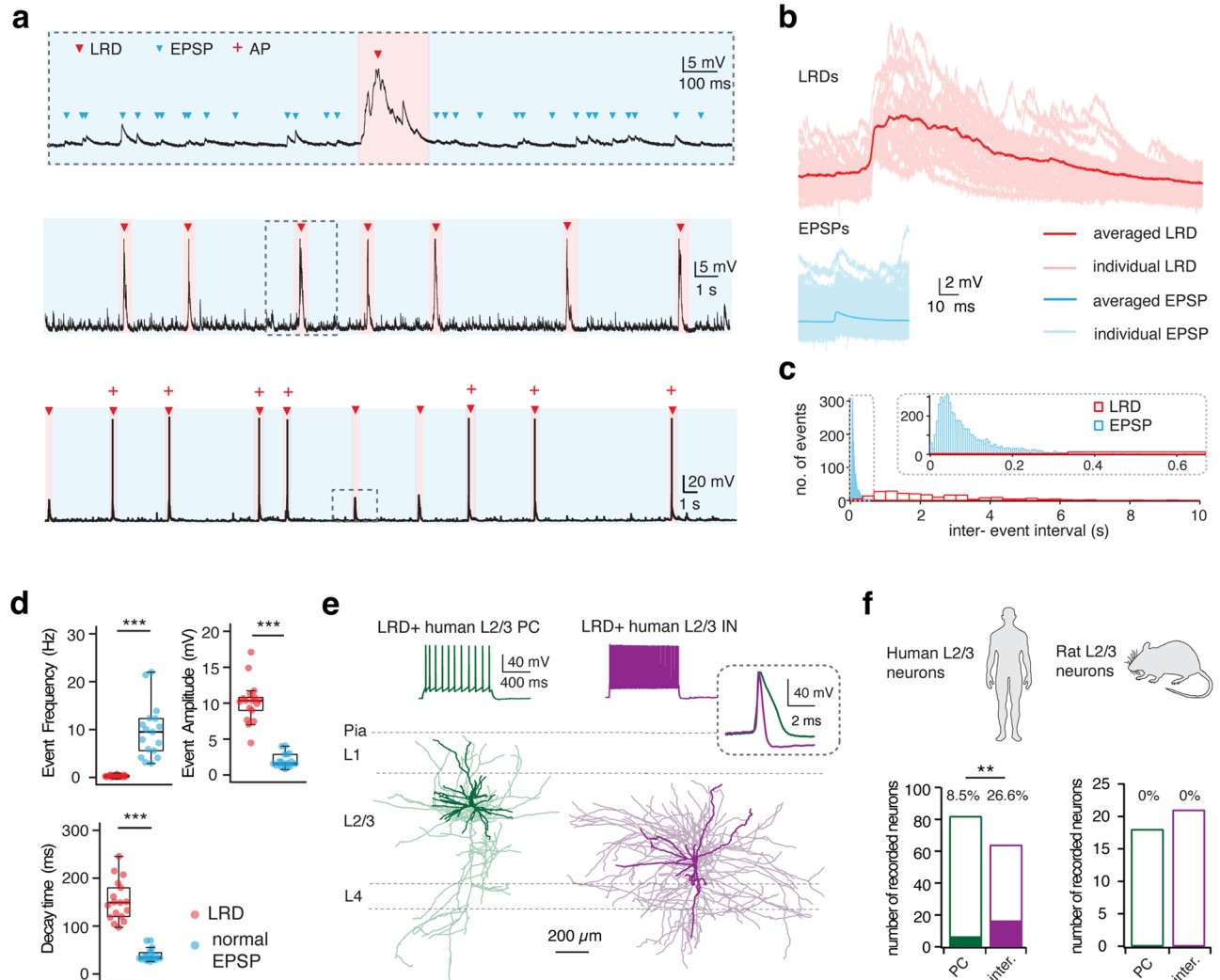

**Fig. 1 | LRDs are identified in a subset of human neocortical L2/3 neurons.**
**a** Whole-cell intracellular recordings from a L2/3 interneuron in the human neo-cortex. Top trace: a typical LRD in the temporal resolution of 2 s. The same LRD is shown in the middle trace on an expanded time scale. Excitatory postsynaptic potentials (EPSPs) are marked by blue while LRDs are marked by red arrowheads. Middle trace: continuous 30 s recording of spontaneously, rhythmically occurring LRDs. Bottom trace: spontaneous LRDs triggering action potential discharge. **b** Top: Mean and individual LRDs ($n = 29$) are superimposed and given in dark and light red, respectively. Bottom: Mean and individual EPSPs ($n = 559$) are superimposed and given in dark and light blue, respectively. Events were extracted and analyzed from 100 s continuous recordings. **c** A 300 s recording was obtained from the same

neuron in (**a**, **b**) and interval histograms of events are shown. The histograms of LRDs and EPSPs were constructed with 33 ms and 5 ms bins, respectively. Inset, EPSP histogram at an expanded time scale. **d** Box plots comparing event frequency, amplitude and decay time for LRDs and EPSPs. $n = 17$ neurons for each group; ***$P < 0.001$ for the Wilcoxon Mann–Whitney U test. **e** Top: Representative firing patterns of a human L2/3 PC and an interneuron exhibiting LRDs. The inset shows the first AP elicited by rheobase current at high temporal resolution. Bottom: Corresponding morphological reconstructions of the neurons shown above. The somatodendritic domain is given in a darker, the axons in a lighter shade. **f** Number of LRD+ and LRD- neurons in L2/3 of human and rat cortex. Numbers above bar graphs indicate the percentage of LRD+ neurons. $P = 0.0035$ for $\chi^2$ test.

human neocortex is more predisposed to produce LRDs than the rodent neocortex, likely due to the notably strong excitatory-to-inhibitory connectivity present in the human L2/3 neocortex[28,33].

It has been reported that in vitro slow oscillations can be induced in ferret visual and prefrontal cortices when perfused with a high K⁺/low Ca²⁺ ACSF 13. We performed intracellular recordings in L2/3 human neurons using an ACSF containing 3.5 mM $K^+$/1 mM $Ca^{2+}$. In 5 out of 7 neurons we identified typical 'Up' states comprised a mix of inhibitory and excitatory inputs (Supplementary Fig. S5a). These 'Up' states had durations in the second range and occurred less frequently than LRDs ($0.02 \pm 0.01$ vs. $0.32 \pm 0.18$ Hz, $P < 0.001$) (Supplementary Fig. S5b1, b2, d). In one PC we observed the simultaneous occurrence of LRDs and 'Up' states reinforcing the notion that LRDs signify a unique and novel type of network behavior (Supplementary Fig. S5b3, c). A detailed comparison between LRDs and 'Up' states is presented in Supplementary Fig. S5d.

## Interneurons with LRD are large basket cells

Neocortical interneurons can be divided into several distinct subtypes based on their electrophysiological, morphological and transcriptional properties[34–37]. To investigate whether LRDs were observed in a specific interneuron subtype, we analyzed the electrophysiological properties and performed 3D morphological reconstructions of L2/3 interneurons with and without LRDs. No clear correlation between LRD occurrence and neuronal firing pattern was found: both LRD+ and LRD- interneurons exhibit diverse firing patterns comprising typical fast-spiking (FS) and non-fast spiking (nFS, including adapting, late spiking, etc.) (Fig. 2i; Supplementary Tab. S2)[38,39]. However, after detailed analysis of passive and active properties, LRD+ interneurons were found to have a smaller input resistance compared to LRD- interneurons ($192.4 \pm 85.1$ vs. $298.5 \pm 117.9$ MΩ, $P < 0.05$). We next studied the morphology of LRD+ and LRD- L2/3 interneurons. Notably, LRD+ interneurons were mostly large basket cells

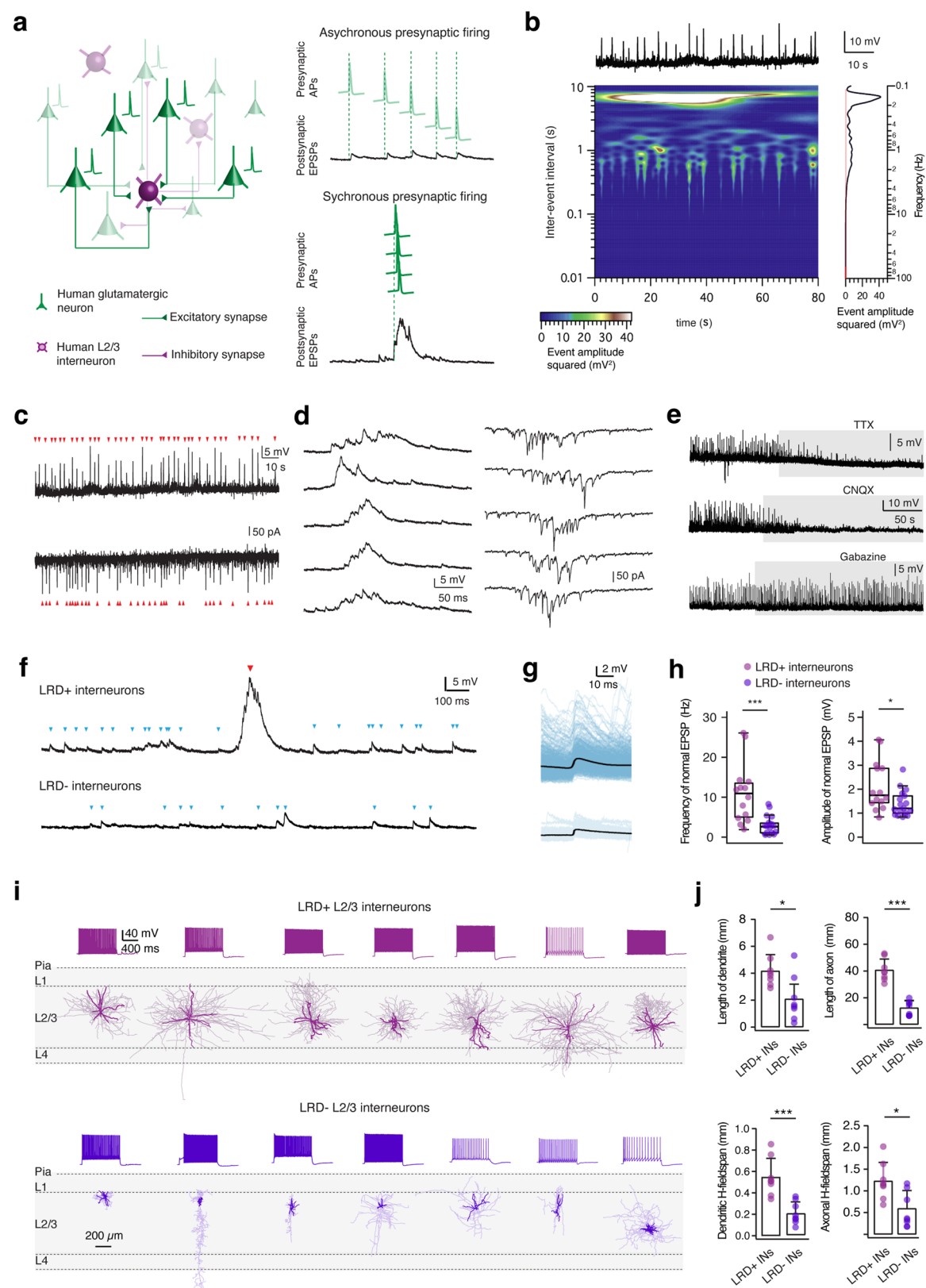

with dense and broad dendritic and axonal domains. In contrast, LRD- interneurons comprised multiple morphological subtypes including small basket, bipolar, double bouquet, and neurogliaform cells (Fig. 2i)[37]. LRD+ interneurons showed significantly longer dendrites and axons than LRD- interneurons (4.2 ± 1.2 vs. 2.1 ± 1.1 mm for dendritic length, $P < 0.05$;

40.8 ± 8.2 vs. 12.5 ± 5.4 mm for axonal length, $P < 0.001$). The horizontal dendritic and axonal fieldspan was wider for LRD+ than for LRD- interneurons (546 ± 176 μm vs. 212 ± 104 μm for dendrites and 1228 ± 427 μm vs. 604 ± 405 μm for axons, respectively; Fig. 2j). Moreover, a larger vertical dendritic fieldspan was also observed for LRD+ neurons (652 ± 194 vs.

**Fig. 2 | LRDs are complex network events depending on glutamatergic transmission and human L2/3 interneurons showing LRDs are large basket cells.**
**a** Left: Synaptic wiring scheme between human L2/3 interneurons and cortical PCs. Right: Diagram summarizing the possible mechanism of the generation of EPSPs and a LRD. Presynaptic asynchronous APs are shown in light green while synchronous APs are in dark green. **b** Time-frequency representation of excitatory spontaneous activities in a LRD+ human L2/3 interneuron. Top plot: Original current clamp recordings in time from a human L2/3 interneuron. Central plot: A time-frequency decomposition of the recording shown above. Squared event amplitude is depicted by heat map colors. Right plot: Amplitude spectrum of excitatory spontaneous activities. The spectrum is shown vertically in frequency (0.05–100 Hz), the red line represents baseline noise. **c** Continuous current (top) and voltage clamp (bottom) recordings of the same LRD+ neuron. **d** Left, five consecutive LRDs recorded in current clamp mode. Right, five consecutive large rhythmic network activities recorded in voltage clamp mode in the same LRD+ neuron sh. **e** Representative current clamp recordings showing block of LRDs in human L2/3 interneurons by TTX (0.5 μM, top trace) and CNQX (10 μM, middle trace) but no effect of gabazine (1 μM, bottom trace) on LRD frequency. **f** Representative continuous recordings from LRD+ (top trace) and LRD- (bottom trace) human L2/3 interneurons. EPSPs are marked in blue while LRDs are marked in red. **g** EPSPs were extracted and analyzed from the same LRD+ interneuron (top) and LRD- interneuron (bottom) in (**d**). The average and individual EPSP traces are superimposed and given in black and blue, respectively. **h** Box plots comparing the frequency and amplitude of EPSPs for LRD+ and LRD- interneurons. $n = 14$ for LRD+ interneurons and $n = 17$ for LRD- interneurons. *$P < 0.05$, ***$P < 0.001$ for the Wilcoxon Mann–Whitney U test. **i** Representative morphological reconstructions and the corresponding firing patterns of seven LRD+ (top) and seven LRD- (bottom) interneurons. The somatodendritic domain is shown in black and axons are shown in gray. **j** Histograms comparing several morphological properties of LRD+ and LRD- human L2/3 interneurons. $n = 8$ for each group. *$P < 0.05$, ***$P < 0.001$ for the Wilcoxon Mann–Whitney U test.

$308 \pm 217$ μm, $P < 0.01$). More details regarding morphological properties and statistical comparisons are given in Supplementary Tab. S2.

Parvalbumin (PV)-expressing GABAergic interneurons are known to exert perisomatic inhibition onto PCs and contribute to cortical network oscillations[40]. A large proportion of basket cells in the human neocortex are PV-expressing neurons displaying a FS firing pattern[41,42]. To identify the expression of PV in human L2/3 interneurons, we performed whole-cell recordings with simultaneous filling of biocytin and the biocytin-conjugated fluorescent Alexa Fluor 594 dye and subsequent immunolabelling. We found that FS interneurons showing LRDs were PV-positive, while nFS interneurons showing LRDs were PV-negative (Supplementary Fig. S6). This indicates that, although LRD+ interneurons show uniform morphologies, they contain more than one transcriptional type of basket cells including FS PV-positive cells and also nFS, possibly cholecystokinin (CCK)-expressing interneurons[43,44].

### Norepinephrine (NE) induces LRDs or enhances their frequency via β-adrenergic receptors
In a subset of human L2/3 interneurons ($n = 7$), LRDs were induced following bath-application of 30 μM NE (Supplementary Fig. S7). The morphological and functional properties of these neurons suggest that they are large basket cells exhibiting high-frequency background EPSPs, similar to those L2/3 interneurons with spontaneous LRDs under control conditions. Thus, they were included in the statistical analysis of electrophysiological and morphological properties of LRD+ interneurons in Fig. 2 and Supplementary Tab. S2. Furthermore, in L2/3 interneurons with spontaneous LRDs, 30 μM NE significantly increased LRD frequency. The NE-induced changes in LRD frequency were reversible following washout (Fig. 3a). A similar increase of the LRD frequency was also observed in the presence of only 10 μM NE (Supplementary Fig. S8). To examine whether NE specifically altered the intrinsic membrane properties in LRD+ interneurons, we measured NE-induced changes in the resting membrane potential ($V_m$) but found no significant effect in both LRD+ and LRD- interneurons (Fig. 3b).

LRDs and background EPSPs were analyzed separately for control conditions and in the presence of 30 μM NE (Fig. 3c). Our results demonstrate that NE differentially modulates LRDs and EPSPs in LRD+ human L2/3 interneurons. Spectral analysis of the spontaneous activity in LRD+ interneurons revealed that NE significantly increased the frequency of the LRDs (from $0.13 \pm 0.17$ Hz to $0.39 \pm 0.23$ Hz, $n = 12$ neurons, $P < 0.001$) without affecting the frequency of the EPSPs ($10.16 \pm 8.53$ vs. $10.86 \pm 6.52$ Hz, $P = 0.5796$, Fig. 3d). In addition, while NE slightly reduced the amplitude of LRDs (from $11.92 \pm 4.07$ to $10.02 \pm 3.53$ mV, $P < 0.05$), it increased that of EPSPs (from $1.69 \pm 0.65$ to $2.11 \pm 1.02$ mV, $P < 0.05$) (Fig. 3e, f). To determine the specific adrenergic receptor type mediating the NE-induced increase in LRD frequency, we applied either 2 μM prazosin (an α1-adrenergic receptor antagonist) or 20 μM propranolol (a β-adrenergic receptor antagonist) together with 30 μM NE, following a bath-application of NE alone. While prazosin had no effect on the adrenergic response, propranolol completely blocked the NE effect on LRD frequency. The LRD frequency increased from $0.11 \pm 0.09$ to $0.28 \pm 0.04$ Hz during the NE application and returned to control level ($0.09 \pm 0.10$ Hz) following co-application of NE and propranolol (Fig. 3g, h). These experiments indicate that the increase in LRD frequency due to NE is mediated by β-adrenoreceptors activation.

### Effect of NE on Global cortical network activity
To assess the effect of NE on overall cortical network activity, we made recordings in human cortical brain slice cultures using a 256 channel multielectrode array (MEA)) both before and after the bath-application of NE and ACh. Under control conditions the analyzed slices ($n = 25$, 6–16 days in vitro (DIV)) exhibited spontaneous network activity (Figs. 3i, j and 4I, j) with an average firing rate across all channels of $10.5 \pm 15.4$ Hz. There were $23.4 \pm 42.7$ (out of 256) active channels and $5.7 \pm 8.6$ channels with detected local field potentials (LFPs; see methods for details). As a summated signal of concurrent neuronal activity close to the recording site, LFP illustrates the dynamic progression of network activity. The average amplitude of the negative LFP was $-57.8 \pm 46.7$ μV. The bath application of 30 μM NE ($n = 13$) resulted in a marked increase in the global firing rate (from $10.5 \pm 17.2$ Hz to $14.8 \pm 12.8$ Hz, *$p < 0.05$), paired with a significant surge in the negative amplitude of the detected LFPs (from $-53.2 \pm 45.5$ μV to $-76.3 \pm 59.4$ μV, *$p < 0.05$) (Fig. 3j–l).

We used graph analysis to quantify the synchronicity degree of neuronal network activity, calculating the degree of centrality for channels with simultaneous spiking activity as a surrogate measure for synchronicity. The mean degree centrality (MDC) for slices treated with NE rose from $0.09 \pm 0.09$ to $0.16 \pm 0.15$ (*$p < 0.05$, Fig. 3l and Supplementary Fig. S9e). The increased activity was mainly noted at electrodes that were already active under control conditions or those nearby (Fig. 3j). No significant difference in the number of active channels with an LFP was observed (Fig. 3l). The increase in firing rate following bath-application of NE was statistically significant for electrodes located in both L2/3 and L5/6.

### Acetylcholine (ACh) suppresses LRDs via the activation of M4Rs
ACh shifts cortical dynamics from a synchronous to asynchronous state and improves the signal-to-noise ratio of sensory signaling[19,21]. To elucidate cholinergic effects on human L2/3 interneurons, 30 μM ACh was bath applied and ACh-induced changes in $V_m$ were compared between interneurons with and without LRDs. ACh persistently depolarized all tested human L2/3 interneurons, an effect that was reversible on washout with ACSF. LRD+ and LRD- L2/3 interneurons were depolarized by $4.0 \pm 3.5$ mV ($n = 10$) and $3.1 \pm 2.6$ mV ($n = 12$), respectively (Fig. 4b). No significant difference in depolarization amplitude was observed.

We found that ACh application resulted in a marked suppression of both spontaneous (Fig. 4a) and NE-induced LRDs (Supplementary Fig. S7). To systematically study cholinergic modulation of spontaneous activity, we analyzed LRDs and background spontaneous EPSPs from continuous

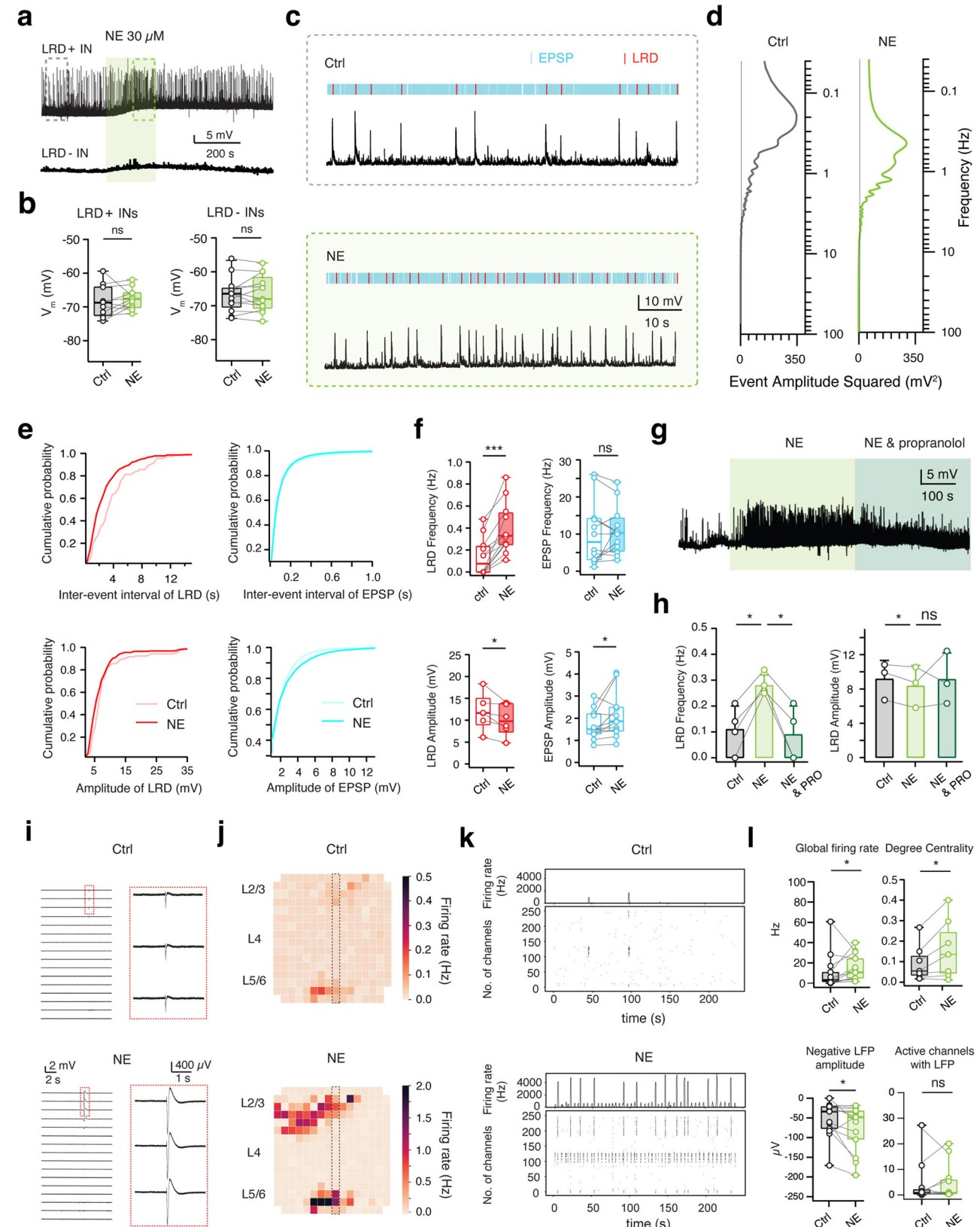

current-clamp recordings during 100 s epochs in each LRD+ interneuron under control conditions and in the presence of ACh (Fig. 4c). Spectral analysis of spontaneous activity in LRD+ interneurons indicates that ACh blocked large amplitude events but increased the frequency of small amplitude events (Fig. 4d). In fact, ACh either decreased ($n = 6$) or

completely blocked ($n = 4$) the occurrence of LRDs, resulting in a reduction of LRD frequency from $0.40 \pm 0.17$ Hz to $0.13 \pm 0.13$ Hz. Of note, ACh at a lower concentration of 10 µM similarly reduced LRD frequency (Supplementary Fig. S8). In contrast, 30 µM ACh significantly increased the frequency of EPSPs from $11.18 \pm 6.74$ Hz to $14.12 \pm 7.99$ Hz (Fig. 4e, f). Our

**Fig. 3 | NE increases LRD frequency via activating β-adrenergic receptors.**
**a** Representative LRD+ (LRD + ) interneuron (IN; top trace) and LRD- (LRD-)
interneuron (bottom trace) showing depolarizing responses following the bath
application of 30 μM NE. The area in the dashed boxes is enlarged in (**c**). **b** Summary
plots showing the resting membrane potential ($V_m$) of human L2/3 interneurons
under control conditionsand in the presence of NE ($n = 10$ for LRD+ interneurons
and $n = 12$ for LRD-interneurons). ns (not significant); Wilcoxon signed-rank test.
**c** 80 s recording from the same LRD+ interneuron under control (top) and in the
presence of NE (bottom). Excitatory postsynaptic potentials (EPSPs) are marked in
blue while LRDs are marked in red. **d** Amplitude spectrum of excitatory spontaneous
activities analyzed from recording traces in (**c**). The spectrums are shown vertically
in frequency (0.05–100 Hz) and the black lines represent baseline noise.
**e** Cumulative distributions of inter-event intervals and amplitudes of excitatory
spontaneous activity recorded in LRD+ human L2/3 interneurons under control
and NE conditions. **f** Box plots summarizing the NE effect on frequency and
amplitude of LRDs and EPSPs in LRD + L2/3 human interneurons ($n = 12$). ns (not
significant), $*P < 0.05$, $***P < 0.001$; Wilcoxon signed-rank test. **g** Representative

current-clamp recordings, following the bath application of NE, showing an increase
of LRD frequency in a human L2/3 interneuron. The effect is blocked by the β-
adrenoreceptor antagonist propranolol (20 μM). **h** Summary histograms of LRD
frequency and amplitude under control, NE and propranolol conditions. ns (not
significant), $*P < 0.05$ for paired student t-test. **i** Representative voltage traces from
16 channels of the MEA under control conditions (top) and 30 μM NE (bottom)
showing an increase in LFP amplitude. Red insets show an enlarged view of the same
three electrodes in both conditions. **j** Heatmap of the average firing rate over the
MEA grid from a five-minute recording in control condition and in the presence of
NE showing an increase synchronous firing in L2/3 and deep layers. **k** Raster plots of
the detected APs over a five-minute recording period under control condition and in
the presence of NE showing an increase in synchronous firing. **l** Box plots displaying
the group effects of NE compared to control on the global firing rate ($n = 13$), degree
of centrality ($n = 7$), the average negative amplitude of the LFPs ($n = 13$) and the
number of channels with detected LFPs ($n = 13$). Ns (not significant), $*P < 0.05$ for
Wilcoxon signed-rank test.

---

data suggests that ACh modulates unitary synaptic and LRD activity in a
distinctive fashion, leading to a desynchronization of synaptic potentials
thereby reducing LRD frequency or abolishing LRD occurrence alto-
gether. Additionally, ACh slightly decreased LRD amplitude from
$12.74 \pm 2.94$ mV to $10.37 \pm 4.24$ mV but had no effect on the amplitude
of EPSPs ($1.78 \pm 0.80$ vs. $1.78 \pm 0.65$ mV, $P = 0.9856$) (Fig. 4e, f). It has
been reported that by blocking $K^+$ conductances, ACh terminates 'Up'
states via muscarinic receptors[20,45]. Here we found that the cholinergic
modulation of LRDs was blocked by administration of a muscarinic $M_4$
receptor ($M_4R$) antagonist, tropicamide, but not by mecamylamine, a
general nicotinic antagonist. The LRD frequency, which was reduced by
ACh, increased to control level in the presence of 1 μM tropicamide
(from $0.07 \pm 0.06$ to $0.28 \pm 0.13$, $P < 0.05$). Likewise, the ACh-induced
decrease of the LRD amplitude was reversed as well (from $6.44 \pm 3.65$ to
$8.16 \pm 3.54$ mV, $P < 0.05$, Fig. 4g, h). These results suggest that ACh
suppresses LRDs mainly via the activation of $M_4Rs$.

### Effect of ACh on cortical network activity

To further understand how ACh influences cortical network activity, we
assessed its effect in human cortical slice cultures using the MEA recording
system. Bath application of ACh (15–30 μM, $n = 12$) resulted in a substantial
increase in the average AP firing rate (from $9.7 \pm 13.7$ Hz to $50.1 \pm 62.2$ Hz,
$**p < 0.01$, Fig. 4l). Yet, this increased firing rate appeared patchy and dis-
junct (Fig. 4j). Additionally, the temporal and spatial correlation observed
under control conditions was disrupted in the presence of ACh (Fig. 4j, k).
Moreover, the rise in firing rate did not coincide with an increase in LFP
amplitude (which would represent synchronous network events). Instead,
there was a significant reduction in the number of active electrodes showing
LFPs (Fig. 4l). The graph analysis to quantify the synchronicity degree of
neuronal network activity revealed a drop in the degree of centrality from
$0.17 \pm 0.18$ to $0.12 \pm 0.11$ ($*P < 0.05$) after ACh application (Fig. 4l and
Supplementary Fig. S9f). We observed no notable difference between the
firing rate increase in electrodes situated in L2/3 or L5/6 following the ACh
application. To further probe how AP firing was modulated by NE and ACh,
30 μM NE and 30 μM ACh were applied on L2/3 interneurons showing
LRD-induced AP firing. We found that NE increased the LRD frequency
and in turn enhanced the LRD-induced AP firing rate. In contrast, appli-
cation of ACh strongly depolarized neurons and thus increase the overall AP
firing rate. However, AP firing was no longer correlated, thereby preventing
the generation of LRDs (Supplementary Fig. S10). This indicates that NE
and ACh enhance AP firing rates in L2/3 interneurons through distinct
mechanisms.

### Discussion

In this study, we uncovered and characterized a rhythmic network event,
revealing cell type-specific network activity in layer 2/3 of human neocortex,
here coined large rhythmic depolarization (LRD). LRDs appeared in a low

frequency range (0.1–0.7 Hz), displayed large amplitudes, long decay times
and sometimes triggered AP firing. Although L2/3 PCs occasionally showed
LRDs, they were predominantly observed in a subset of L2/3 interneurons
exhibiting dendritic and axonal morphologies similar to those of large
basket cells[46]. In contrast to human neocortex, LRDs were not observed in
layer 2/3 of rat frontal or temporal cortex under identical recording con-
ditions. Our data suggest that LRDs are triggered by near-synchronous
presynaptic AP firing in glutamatergic neurons. Furthermore, NE and ACh
differentially modulate synchronous network events and asynchronous
unitary synaptic inputs. NE increased LRD frequency via β-adrenergic
receptors activation without affecting the frequency of EPSPs. Conversely,
ACh decreased LRD frequency via activation of $M_4$ muscarinic receptors
but increased EPSP frequency. Data obtained from MEA recordings further
demonstrated that NE enhanced near-synchronous AP firing whereas ACh
desynchronized network activity. The differential modulation of such
activity by NE and ACh suggests specific modulatory mechanisms in the
human neocortex and sheds light on mechanisms of synchronized neuronal
activity in the human neocortex which is associated with different beha-
vioral states.

'Giant depolarizing potentials (GDPs)' have been described as
network-driven synaptic events in the immature rodent hippocampus and
neocortex. Their initiation requires excitatory GABAergic transmission
which promotes voltage-dependent AP bursts in immature pyramidal
neurons[32,47]. The LRDs discovered and characterized in the present work
were recorded in L2/3 of human neocortex and are dependent on gluta-
matergic transmission and not affected by gabazine, application suggesting a
distinct mechanism in adult human neocortex. In the acute ferret, mouse
and rat neocortex, rhythmic network events can be triggered by perfusing
ACSF containing a high $K^+$/low $Ca^{2+}$ ion concentration. These events have
been characterized as slow oscillations that persist in vitro[13–15]. When this
modified ACSF was applied, 'Up' states, with both depolarizing and
hyperpolarizing components, were observed in L2/3 neurons of acute
human slices. Contrary to LRDs, which were identified in a limited subset of
neurons, 'Up' states were induced in the majority of the recorded neurons.
The duration of these 'Up' states lasted several seconds, noticeably longer
than LRDs, consistent with earlier studies[13–15]. Spontaneous sharp
waves similar to the time course of LRDs, were observed in neocortical slices from
both epileptic[5,6,9] and non-epileptic tissue[9,12]. However, unlike LRDs, which
are always depolarizing and show only minor contribution of inhibitory
inputs, intracellular recordings of neurons involved in spontaneous sharp
wave generation revealed a wider range of potentials, including depolariz-
ing, hyperpolarizing, or a combination of both[5,6,9]. In addition, spontaneous
sharp waves were effectively suppressed by blockade of $GABA_A$ receptors,
whereas LRDs remained unaffected. To our knowledge, the LRDs described
in this study appear to represent a novel form of brain oscillation, as they
show significant differences from previously reported in vitro network
activity.

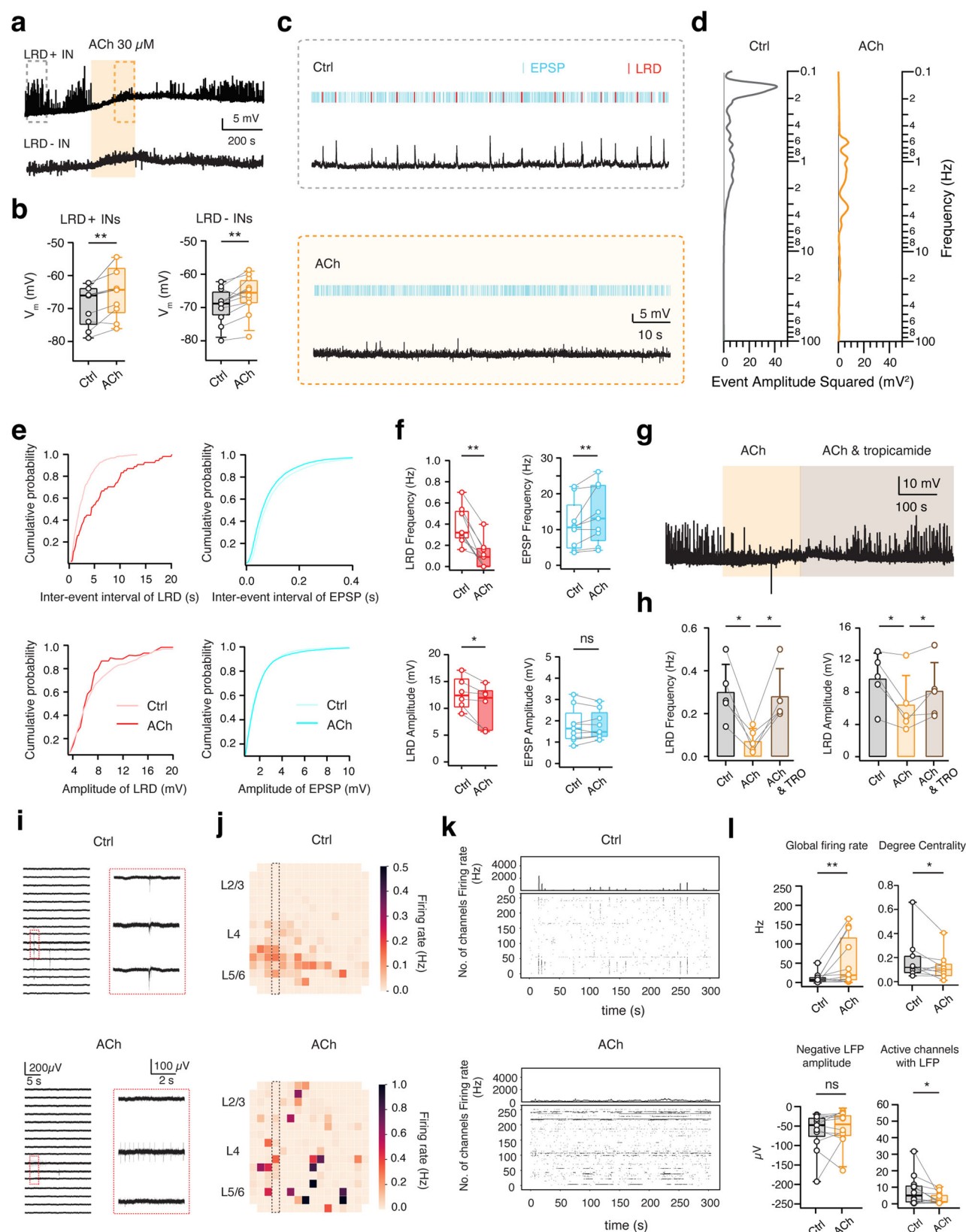

In this study, the occurrence of LRDs showed no correlation with patient age, gender, specific neocortical subregion. The cases were carefully selected, so that the surgical approach tissue was considered according to anatomical, microscopic, imaging, and neuropathological criteria. We acquired spared human cortex samples generated during the surgical approach of patients undergoing operations due to epilepsy or brain tumors. It worth noting that the majority of patients diagnosed with epilepsy or tumor exhibited preoperative seizures. In our experiments, we exclusively used brain tissue located more than 10 mm away from the lesion to ensure the non-pathological nature of the samples. However, the possibility

**Fig. 4 | ACh suppresses LRDs via activating M₄Rs. a** Representative LRD+ interneuron (top trace) and LRD- interneuron (bottom trace) showing depolarizing responses following bath application of 30 μM ACh. The area in the dashed boxes is enlarged in (**c**). **b** Summary plots showing the resting membrane potential (V_m) under control conditions and in the presence of ACh in human L2/3 interneurons. *n* = 10 for each group. **P < 0.01; Wilcoxon signed-rank test. **c** A 80 s recording from a LRD+ interneuron under control (top) and ACh conditions (bottom), respectively. Spontaneous EPSPs are marked in blue while LRDs are marked in red. **d** Amplitude spectra of excitatory spontaneous activity analyzed from recording traces in c. Spectra are shown vertically in frequency (0.05–100 Hz), black lines represent baseline noise. **e** Cumulative distributions of inter-event interval and amplitude of spontaneous excitatory activity in LRD+ human L2/3 interneurons under control conditions and in the presence of ACh. **f** Box plots summarizing the ACh effect on frequency and amplitude of LRDs and EPSPs in LRD + L2/3 human interneurons (*n* = 10). ns (not significant), *P < 0.05, **P < 0.01; Wilcoxon signed-rank test. **g** Representative current-clamp recordings with bath application of ACh

showing a decrease of LRD frequency in a human L2/3 interneuron. The effect is reversed by 1 μM tropicamide. **h** Summary histograms of LRD frequency and amplitude under control conditions, in the presence ACh alone and together with the M₄ mAChR antagonist tropicamide. *P < 0.05; Wilcoxon signed-rank test. **i** Representative voltage traces from 16 channels of the MEA under control conditions (top) and 30 μM ACh (bottom) showing block of LFPs by ACh. Red inserts show a detailed view of the same three electrodes in both conditions; note the switch from a synchronized firing pattern under control condition to a desynchronized pattern in the presence of ACh. **j** Heatmap of the average firing rate over the MEA grid from a five-minute recording in control condition and in the presence of ACh showing a decrease in temporal and spatial correlation of global AP firing. **k** Raster plots of the detected action potentials over the five-minute recording period under control **l** Box plots displaying the group effects of ACh compared to control on the global firing rate (*n* = 12), degree of centrality (*n* = 10), the average negative amplitude of the LFPs (*n* = 12) and the number of channels with detected LFPs (*n* = 12). *P < 0.05, **P < 0.01 for Wilcoxon signed-rank test.

remains that these tissues may had been affected by the widespread of seizures prior to surgery. Interpreting LRD as epileptiform-like activity is difficult, as LRD consists solely of EPSPs without IPSP inputs, whereas typical epileptic activity involves both[48]. Nevertheless, the excitatory-to-inhibitory balance of the circuitry in resected human tissue might have been altered by preoperative seizures, potentially facilitating the generation of rhythmic network activity. While we did observe LRDs in a patient without preoperative seizures, further investigation with additional seizure-free samples is required to conclusively demonstrate that LRDs are unrelated to seizure attacks.

Our preliminary recordings of L2/3 neurons in peritumoral area of human brain slices showed that neurons in this area displayed high input resistance, low rheobase current and discontinued AP firing when compared with neurons recorded in tumor-free healthy tissue. Neurons included in our dataset neither showed abnormalities in firing patterns nor in other electrophysiological nor in morphological properties when compared to previous description of human cortical neurons[49–52]. LRDs were only observed within a short time window (6 h) after slice preparation. In this study, we were able to begin preparation of neocortical slices within 10 min after resection of human tissue. This is likely to contribute to the maintenance of the rhythmic network activity in acute human brain slices studied here.

The diversity of cortical interneurons has posed a major challenge to classify and characterize their defining properties. Recent research on the human brain has shed light on this complexity by mapping the transcriptomically defined cell types within human cortical taxonomy[37,53]. However, the synaptic activity were not investigated in these studies, as they used a recording ACSF that contained blockers for both glutamatergic and GABAergic receptors. Here, we described for the first time LRDs that were frequently observed in a specific morphological type of interneurons, namely large basket cells. Large basket cells showing LRDs display more frequent and larger EPSPs, resulting in a greater membrane depolarization and in turn a higher probability of the neuron reaching the threshold for AP firing. However, these interneurons show a substantially lower input resistance compared to those without LRD activity, indicating that they are less excitable and require a higher current influx to generate APs. It is therefore likely that the synchronous network events, LRDs, are critical for eliciting APs in these large basket cells and in turn trigger feedforward/feedback inhibition of connected postsynaptic neurons. Previous studies showed that rat PV-positive basket cells have a low input resistance[44,54,55]. By providing perisomatic inhibition to pyramidal cells, PV basket cells have been implicated in the generation of gamma oscillations (30–120 Hz) through feedforward and feedback inhibition[1,56]. Apart from this, inhibition of PV-positive interneurons disrupts learning-induced augmentation of delta (0.5–4 Hz) and theta oscillations (4–10 Hz) in mouse hippocampus[57]. During slow oscillations (< 1 Hz), PV-interneurons were the most active interneuron subtype during cortical 'Up' states in rodent neocortex 15.

Apart from PV-positive chandelier cells, many PV-expressing interneurons show 'basket cell'-like morphology 41. We found that LRD+ interneurons not only comprise PV-positive FS interneurons but also nFS interneurons which do not express PV. It is conceivable that these interneurons are CCK-positive interneurons with an nFS firing pattern. Although they target the same somatodendritic compartments of pyramidal cells as PV-expressing interneurons, CCK- and PV-positive interneurons play complementary roles in network oscillations[58,59]. In mouse prefrontal and temporal cortex, PV- and CCK-expressing neurons constitute approximately 30–40% of all GABAergic interneurons, a percentage similar to the LRD+ neurons we recorded among all the interneurons[60,61]. Interneurons showing LRDs have significantly broader and longer dendrites and axons than LRD- interneurons. The broad and dense axonal arborizations of PV- and CCK- basket cells makes them a powerful inhibitory force in human layer 2/3 and may therefore play a dominant role in synchronized network oscillations.

We propose that the initiation of LRDs is driven by a summation of near-synchronous unitary EPSPs. Neurons that show LRDs were found to display a larger mean EPSP amplitude and higher frequency of spontaneous unitary EPSPs. With higher EPSP amplitude and/or more synapses converging on a specific postsynaptic neuron, EPSP summation to a critical level and hence the initiation of LRDs becomes more likely. This could be a contributing factor why LRDs were not observed in rat neocortex as human pyramidal cells establish stronger and more reliable synaptic connections in the local neuronal circuitry[33]. Recent studies revealed that human cortex has a much higher fraction of interneurons (approximately 2.5-fold) than rodent neocortex[62]. Considering LRDs are more prominently observed in interneurons rather than PCs, human neocortex is therefore more prone to generate synchronous network activity than rodent neocortex under the same conditions.

In this study we demonstrated for the first time that NE can initiate or increase the frequency of rhythmic brain oscillations. This adrenergic modulation of LRDs may be attributed to several potential mechanisms. NE may increase the EPSP amplitude in L2/3 large basket cells of human neocortex which in turn thus promoting the emergence of LRDs. NE might potentially trigger persistent firing by activating α₂ receptor-gated HCN channels which could contribute in the synchronization of neuronal activity as demonstrated in layer 2/3 of rat prefrontal cortex[63]. However, this is in contrast to the present finding that the NE-induced enhancement of LRD frequency is mediated by β-adrenergic receptor activation.

It has been reported that the activation of muscarinic ACh receptors by stimulating the brainstem cholinergic nucleus abolishes slow oscillations[20,45]. We hypothesize that through activation of G_s protein-coupled β-adrenergic receptors, NE regulates LRD frequency in a way opposite to ACh via modulation of Ca²⁺ and K⁺ conductance. In accordance with this hypothesis, we found that ACh can block either spontaneous or NE-induced LRDs by activating G_{i/o} protein-coupled M₄Rs. Moreover, ACh increased the overall frequency of unitary EPSPs, possibly via nAChR activation,

suggesting that ACh causes desynchronization of LRDs into unitary EPSPs and a decorrelation of responses between neurons. In summary, LRD generation was promoted or suppressed by NE and ACh, respectively. The adrenergic and cholinergic modulation of LRDs drives temporal dynamics of cortical activity and controls cortical information processing and transitions between brain states.

## Conclusions

To our knowledge, the study discovered and characterized a new form of rhythmic network activity in the human neocortex called Large Rhythmic Depolarizations (LRDs), predominantly observed in a subset of L2/3 interneurons. Unlike previously described activities, LRDs are independent of GABA$_A$ receptors, relying instead solely on glutamatergic transmission; they demonstrate distinct features from other known in vitro network activities. While prevalent in human neocortex samples, LRDs were notably absent in rat frontal or temporal cortex. Modulation studies revealed distinct and differential impacts of norepinephrine (NE) and acetylcholine (ACh) on synchronous and asynchronous network events, offering insights into specific modulatory mechanisms in the human neocortex.

## Methods

### Patients and animals

All patients underwent neurosurgical resections because of pharmaco-resistant epilepsy or tumor removal. Written informed consent to use spare neocortical tissue acquired during the surgical approach was obtained from all patients. The study was reviewed and approved by the local ethic committee (EK067/20). All ethical regulations relevant to human research participants were followed. For this study, we collected data from 21 patients (14 females, 7 males; age ranging from 8 to 75 years old) (Supplementary Tab. S1). The cases were meticulously selected to fulfill two main criteria: 1) availability of spare tissue based on the needed surgical approach; and 2) normal appearance of the tissue according to radiological and intraoperative criteria (absence of edema, absence of necrosis, and sufficient distance to any putative intracerebral lesion). In addition, samples from tumor cases were neuropathologically reviewed to rule out the presence of tumor cells in the examined neocortical specimen.

All experimental procedures involving animals were performed in accordance with the guidelines of the Federation of European Laboratory Animal Science Association, the EU Directive 2010/63/EU, and the German animal welfare law. In this study, Wistar rats (Charles River, either sex) aged 40–55 postnatal days were anesthetized with isoflurane and then decapitated. Rats were obtained from Charles River and kept under a 12 h light–dark cycle, with food and water available ad libitum.

### Slice preparation

Human cortex was carefully micro-dissected and resected with minimal use of bipolar forceps to ensure tissue integrity. Resected neocortical tissue from the temporal or frontal cortex was directly placed in an ice-cold artificial cerebrospinal fluid (ACSF) containing (in mM): 110 choline chloride, 26 NaHCO$_3$, 10 D-glucose, 11.6 Na-ascorbate, 7 MgCl$_2$, 3.1 Na-pyruvate, 2.5 KCl, 1.25 NaH$_2$PO$_4$, and 0.5 CaCl$_2$) (325 mOsm/l, pH 7,45) and transported to the laboratory. Slice preparation commenced within 10 min after tissue resection. The pia was carefully removed from the human tissue block using forceps and the pia-white matter (WM) axis was identified. 300 μm thick slices were prepared using a Leica VT1200 vibratome in ice-cold ACSF solution containing 206 mM sucrose, 2.5 mM KCl, 1.25 mM NaH$_2$PO$_4$, 3 mM MgCl$_2$, 1 mM CaCl$_2$, 25 mM NaHCO$_3$, 12 mM N-acetyl-L-cysteine, and 25 mM glucose (325 mOsm/l, pH 7,45). During slicing, the solution was constantly bubbled with carbogen gas (95% O$_2$ and 5% CO$_2$). After cutting, slices were incubated for 30 min at 31–33 °C and then at room temperature in ACSF containing (in mM): 125 NaCl, 2.5 KCl, 1.25 NaH$_2$PO$_4$, 1 MgCl$_2$, 2 CaCl$_2$, 25 NaHCO$_3$, 25 D-glucose, 3 myo-inositol, 2 sodium pyruvate, and 0.4 ascorbate (300 mOsm/l; 95% O$_2$ and 5% CO$_2$). To maintain adequate oxygenation and a physiological pH level, slices were kept in carbogenated ACSF (95% O$_2$ and 5% CO$_2$) during the transportation.

Rat brains were quickly removed and placed in an ice-cold sucrose containing ACSF. The experimental procedures used here have been described in detail previously[64]. 300 μm thick coronal slices of the prelimbic medial prefrontal cortex (mPFC) and temporal association cortex were cut and incubated using the same procedures and solutions as described above for human slices.

### Organotypic slice cultures of human neocortex

Preparation and cultivation of slice cultures of human neocortex followed previously published protocols 65. In brief, the neocortex was carefully micro-dissected and resected with only minimal use of bipolar forceps to ensure tissue integrity, directly transferred into ice-cold artificial cerebrospinal fluid (ACSF) (in mM: 110 choline chloride, 26 NaHCO$_3$, 10 D-glucose, 11.6 Na-ascorbate, 7 MgCl$_2$, 3.1 Na-pyruvate, 2.5 KCl, 1.25 NaH$_2$PO$_4$, and 0.5 CaCl$_2$) equilibrated with carbogen (95% O$_2$, 5% CO$_2$) and immediately transported to the laboratory. Tissue was kept always submerged in cool and carbogenated ACSF. After removal of the pia, tissue blocks were trimmed perpendicular to the cortical surface and 250 μm thick slices were prepared using a live tissue vibratome. After the cortical tissue was sliced as described above, slices were cut into several evenly sized pieces. Subsequently, slices were transferred onto culture membranes (uncoated 30 mm Millicell-CM tissue culture inserts with 0.4 μm pores, Millipore) and kept in six-well culture dishes (BD Biosciences). For the first hour following the slicing procedure, slices were cultured on 1.5 ml intermediate step HEPES media (48% DMEM/F-12 (Life Technologies), 48% Neurobasal (Life Technologies), 1x N-2 (Capricorn Scientific), 1x B-27 (Capricorn Scientific), 1x Glutamax (Life Technologies), 1x NEAA (Life Technologies) + 20 mM HEPES before changing to 1.5 ml hCSF per well without any supplements. No antibiotics or antimycotics were used during cultivation. The plates were stored in an incubator (MCO-170AICUVH-PE, PHC Corporation) at 37 °C with 5% CO$_2$ and 100% humidity. For MEA recordings, slice cultures were transferred into the recording chamber of a MEA Setup (described below).

### Whole-cell recordings

Whole cell recordings were performed in acute slices 30 hours at most after slice preparation for human brain tissues and within 8 h for rat brains. During whole-cell patch-clamp recordings, human or rat slices were continuously perfused (perfusion speed ∼ 5 ml/min) with ACSF bubbled with carbogen gas and maintained at 30–33 °C. Patch pipettes were pulled from thick wall borosilicate glass capillaries and filled with an internal solution containing (in mM): 135 K-gluconate, 4 KCl, 10 HEPES, 10 phosphocreatine, 4 Mg-ATP, and 0.3 GTP (pH 7.4 with KOH, 290–300 mOsm). Neurons were visualized using either Dodt gradient contrast or infrared differential interference contrast microscopy. Human L2/3 neurons were identified and patched according to their somatic location (300–1200 μm from pia)[65]. In rat acute prelimbic cortical slices, layer 2 is clearly distinguishable as a thin dark band that is densely packed with neuron somata. Layer 3 is about 2–3 times wider than layer 2 and has about the same width as layer 1. According to previous publications, layer 2/3 was located at a depth of 200 to 550 μm from the pia[66]. Putative PCs and interneurons were differentiated on the basis of their intrinsic action potential (AP) firing pattern during recording and after post hoc histological staining also by their morphological appearance.

Whole-cell patch clamp recordings of human or rat L2/3 neurons were made using an EPC10 amplifier (HEKA). During recording, slices were perfused in ACSF at 31–33 °C containing (in mM): 125 NaCl, 2.5 KCl, 1.25 NaH$_2$PO$_4$, 1 MgCl$_2$, 2 CaCl$_2$, 25 NaHCO$_3$, 25 D-glucose, 3 myo-inositol, 2 sodium pyruvate, and 0.4 ascorbic acid. In a subset of experiments designed to induce 'Up' states, slices were perfused in a modified ACSF at 31–33 °C containing (in mM): 125 NaCl, 3.5 KCl, 1.25 NaH$_2$PO$_4$, 1 MgCl$_2$, 1 CaCl$_2$, 25 NaHCO$_3$, 25 D-glucose, 3 myo-inositol, 2 sodium pyruvate, and 0.4 ascorbic acid. Signals were sampled at 10 kHz, filtered at 2.9 kHz using Patchmaster software (HEKA), and later analyzed offline using Igor Pro software (Wavemetrics). Recordings were performed using patch pipettes of

resistance between 5 and 10 MΩ. Biocytin was added to the internal solution at a concentration of 3–5 mg/ml to stain patched neurons. A recording time > 15 min was necessary for an adequate diffusion of biocytin into dendrites and axons of patched cells[67].

## Multi-electrode array (MEA) recordings

To perform the MEA recordings of the human cortical cultures, the brain slice was excised from the insert with the slice still attached to the culturing membrane. Subsequently, the slice was moved to the MEA chamber and placed onto the electrodes of the MEA Chip with the slice surface facing down. For fixation and improved contact with the electrodes, the slice was fixed in place by a weighted, close-meshed harp (ALA-HSG MEA-5BD, Multi Channel Systems MCS GmbH). Slices equilibrated at least 30 min on the chip with constant carbogenated ACSF (same as used for acute slices) perfusion at 30–33 °C before MEA recordings were started. MEA recordings were performed using a 256–MEA (16 × 16 lattice) with electrode diameter of 30 μm and electrode spacing of 200 μm, thus covering a recording area of ~3.2 × 3.2 mm$^2$ (USB-MEA 256-System, Multi Channel Systems MCS GmbH). Recordings with the 256-MEA were performed at a sampling rate of 10–25 kHz using the Multi-Channel Experimenter (Multi Channel Systems MCS GmbH).

## Drug application

NE (10 μM or 30 μM) and ACh (10 μM, 15 μM or 30 μM) were bath applied for 150–300 s through the perfusion system during whole-cell patch clamp or MEA recordings. In a subset of human neurons, propranolol (20 μM), tropicamide (TRO, 1 μM), tetrodotoxin (TTX, 0.5 μM), cyanquixaline (CNQX, 10 μM), gabazine (1 μM), mecamylamine (1 μM) or prazosin (2 μM) were bath applied for 200–600 s to study the underlying pharmacological mechanisms. Drugs were purchased from Sigma-Aldrich or Tocris.

## Histological staining

After recordings, brain slices containing biocytin-filled neurons were fixed for at least 24 h at 4 °C in 100 mM phosphate buffer solution (PBS, pH 7.4) containing 4% paraformaldehyde (PFA). After rinsing several times in 100 mM PBS, slices were treated with 1% H$_2$O$_2$ in PBS for about 20 min to reduce any endogenous peroxidase activity. Slices were rinsed repeatedly with PBS and then incubated in 1% avidin-biotinylated horseradish peroxidase (Vector ABC staining kit, Vector Lab. Inc.) containing 0.1% Triton X-100 for 1 h at room temperature. The reaction was catalyzed using 0.5 mg/ml 3,3-diaminobenzidine (DAB; Sigma-Aldrich) as a chromogen. Subsequently, slices were rinsed with 100 mM PBS, followed by slow dehydration with ethanol in increasing concentrations, and finally in xylene for 2–4 h. After that, slices were embedded using Eukitt medium (Otto Kindler GmbH).

In a subset of experiments, we tried to identify the expression of the molecular marker - a calcium-binding protein parvalbumin (PV) in human layer 2/3 interneurons. To this end, during electrophysiological recordings, Alexa Fluor 594 dye (1:500, Invitrogen) was added to the internal solution for post hoc identification of patched neurons. After recording, slices (30 μm) were fixed with 4% PFA in 100 mM PBS for at least 24 h at 4 °C and then permeabilized in 1% milk powder solution containing 0.5% Triton X-100 and 100 mM PBS. Primary and secondary antibodies were diluted in the permeabilization solution (0.5% Triton X-100 and 100 mM PBS) shortly before the antibody incubation. For single-cell PV staining, slices were incubated overnight with Rabbit-anti-PV primary antibody (1:120, ab11427, Abcam) at 4 °C and then rinsed thoroughly with 100 mM PBS. Subsequently, slices were treated with Donkey-anti-Rabbit Alexa Fluor secondary antibodies (1:400, A21207, Invitrogen) for 2–3 h at room temperature in the dark. After rinsing with 100 mM PBS, the slices were embedded in Fluoromount. Fluorescence images were taken using the Olympus CellSens platform. The position of the patched neurons was identified by the biocytin conjugated Alexa dye so that the expression of PV could be examined in biocytin-stained neurons. After acquiring fluorescent images, slices were incubated in 100 mM PBS overnight and then used for subsequent histological processing as described above.

## Morphological 3D reconstructions

Using NEUROLUCIDA® software (MBF Bioscience, Williston, VT, USA), morphological reconstructions of biocytin filled human layer 2/3 interneurons were made at a magnification of 1000-fold (100-fold oil-immersion objective and 10-fold eyepiece) on an upright microscope. Neurons were selected for reconstruction based on the quality of biocytin labelling when background staining was minimal. Neurons with major truncations due to slicing were excluded. Embedding using Eukitt medium reduced fading of cytoarchitectonic features and enhanced contrast between layers 67. This allowed the reconstruction of different layer borders along with the neuronal reconstructions. Furthermore, the position of soma and layers were confirmed by superimposing the Dodt gradient contrast or differential interference contrast images taken during the recording. The tissue shrinkage was corrected using correction factors of 1.1 in the x–y direction and 2.1 in the z direction 67. Analysis of 3D-reconstructed neurons was done with NEUROEXPLORER® software (MBF Bioscience, Williston, VT, USA).

## Data analysis

**Single-cell recording data analysis.** Custom written macros for Igor Pro 6 (WaveMetrics) were used to analyze the recorded electrophysiological signals. The resting membrane potential ($V_m$) of the neuron was measured directly after breakthrough to establish the whole-cell configuration with no current injection. The input resistance was calculated as the slope of the linear fit to the current–voltage relationship. For the analysis of single spike characteristics such as threshold, amplitude and half-width, a step size increment of 10 pA for current injection was applied to ensure that the AP was elicited very close to its rheobase current. The spike threshold was defined as the point of maximal acceleration of the membrane potential using the second derivative ($d^2V/dt^2$), which is, the time point with the fastest voltage change. The spike amplitude was calculated as the difference in voltage from AP threshold to the peak during depolarization. The spike half-width was determined as the time difference between rising phase and decaying phase of the spike at half-maximum amplitude.

The spontaneous activity was analyzed using the program SpAcAn (https://www.wavemetrics.com/project/SpAcAn). EPSPs and LRDs were distinguished by dramatic differences in event amplitude and decay time. A threshold of 0.2 mV was set manually for detecting EPSP events while a threshold of 3 mV was set for detecting LRDs. Recordings were not filtered to reduce noise before data analysis. When marking EPSPs, small EPSPs distributing in decay phrase but not rising phrase of LRDs were included into analysis. To study oscillatory network activities, we computed time-frequency representations of the signals by performing wavelet analysis using the Time-Frequency Toolkit (https://www.wavemetrics.com/project/TFPlot). Morlet wavelets were used for decomposition of recording signals as they provide an ideal compromise between time and frequency resolution[68].

**MEA data analysis.** MEA recordings were analyzed using custom-written programs in Python, detecting and quantifying the mean firing rate, number of active channels, bursting channels, and network bursts. First, the raw signal was filtered using a band-pass filter (Butterworth 2nd order). The spike identification was performed according to a threshold-based method using median absolute deviation (MAD) / 0.6745 x -5. Signal deviations were detected and aligned to the next minimum of the signal with a 1 ms dead time. The firing rate for each recording was defined as the number of recorded spikes divided by the duration of the recording (in s). Bursting channels were calculated using a modified adaptive network-wide cumulative moving average (CMA) approach in which all the electrodes of one MEA in multiple measurement time points were analyzed together[69].

In short, all inter-spike intervals (ISIs) were calculated and grouped into 5 ms bins for the whole recording. Next, a CMA over the histogram of the bins was calculated and a burst threshold was determined, therefore adapting the detection of bursts to the basic activity of the entire slice 69. The detection sensitivity was further increased by applying a minimum and maximum threshold of 60 ms and 140 ms, respectively. Whenever the threshold was undercut by at least three consecutive spikes, it was defined as a single-channel burst.

For a quantitative analysis of spiking synchronization, we used a graph theoretical approach to identify the degree of centrality of active channels[70–72]. Specifically, we designated MEA contacts as nodes and the shared spiking time as edges to construct a graph representation of the recorded network activity. To construct edges, we grouped spike trains for each channel into 200 ms long bins and defined two channels as connected by an edge only if they both recorded spikes are in the same bin. By using this approach, we were able to examine the functional connectivity and activity patterns in the neuronal network.

We used mean degree centrality as a measure of the connectivity of nodes in the network (Python library NetworkX). Mean degree centrality (MDC) quantifies the average number of edges connecting a node to other nodes in the network. The degree of each node was defined by the number of edges connected to that node, i.e. the number of other MEA contacts that share a spiking time with that contact within a 200-ms bin. Finally, MDC was calculated by summing the degree of all nodes and dividing it by the total number of nodes in the network, which is

$$MDC = \frac{2 \times number\,of\,total\,edges}{number\,of\,total\,nodes}$$

where the factor 2 is introduced because each edge is counted twice (once for each node it connects). By calculating the MDC for the whole MEA recording, we were able to assess the overall connectivity of the network and identify how nodes connectivity (used as a surrogate for synchronicity) changed upon application of NE and ACh compared to baseline.

Detection of LFPs was performed after low-pass filtering the signal of each channel (Butterworth 2nd order with a Nyquist frequency of 0.5 x sampling rate and a cut-off of 100 Hz); as a threshold, standard deviation of the low pass filtered signal multiplicated by three was used. Any deviation above or below this threshold with a minimum duration of 30 ms was defined as a LFP. The respective maximum and minimum deviation was defined as the amplitude of respective LFP.

All Python scripts are available on GitHub page (https://github.com/jonasort/MEA_analysis/tree/main/modified_common_script).

**Statistical analysis**. Data was either presented as box plots ($n \geq 10$) or as bar histograms ($n < 10$). For box plots, the interquartile range (IQR) is shown as box, the range of values within 1.5∗IQR is shown as whiskers and the median is represented by a horizontal line in the box; for bar histograms, the mean ± SD is given. Wilcoxon Mann-Whitney U-test was performed to access the difference between individual clusters. Statistical significance was set at $P < 0.05$, and n indicates the number of neurons/slices analyzed.

**Reporting summary**
Further information on research design is available in the Nature Portfolio Reporting Summary linked to this article.

**Data availability**
The authors declare that the data supporting the findings of this study are available within the paper and its Supplementary Data files. Supplementary Data 1–4 provided with this study are source data for Figs. 1–4, respectively. Should any raw data files be needed in another format they are available from the corresponding author upon reasonable request. Source data are provided with this paper.

**Code availability**
All the custom code used in the study is available from the corresponding author on reasonable request.

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

## Acknowledgements
We would like to thank Werner Hucko and Birgit Gittel for excellent technical assistance. We thank Dr. Karlijn van Aerde for custom-written macros in Igor Pro software. We are grateful for funding support from the European Union's Horizon 2020 Framework Programme for Research and Innovation under the Framework Partnership Agreement No. 650003 (HBP FPA) to D.F., funding from DFG FOR2715, Chan Zuckerberg Initiative DAF (2020-221779) to H.K. and funding from BMBF (German Ministry of Education and Research, project number 031L0260B) to D.D.

## Author contributions
D.F., D.Y., G.Q., and H.K. designed the experiments. D.Y. and G.Q. carried out the patch-clamp recording experiments from human and rat slices and electrophysiological data analysis. D.Y. performed Neurolucida reconstructions and performed morphological analysis. D.D. performed surgeries on human patients. D.Y., A.B., and H.K. prepared acute human brain slices and A.B. prepared the human slice cultures. J.O. and V.W. performed MEA recordings. J.O., H.K., and D.D. analyzed MEA data. D.Y. and D.F. wrote the manuscript with the inputs from all authors. All authors have given approval for the final version of the manuscript.

## Funding

## Competing interests
The authors declare no competing interests.
