## [Peer Review File · Communications Biology]

REVIEWERS' COMMENTS:

Reviewer #1 (Remarks to the Author):

This article by Yang et al. is a revised version of an article I previously reviewed for Nature Communication. This new version is not very different from the previous one, but the authors have made some improvements and corrections that address the main remaining issues as much as possible. As a result, my assessment of the current version remains very similar to my initial assessment:

1) The experiments and data analysis are well conducted, and I have no concerns about the validity and quality of the data. Although not very innovative and classic in its design, this study presents solid and convincing results. In particular, I find the detailed comparison of the physiology and anatomy of LRD+ and LRD- interneurons remarkable, and overall, the results are novel and original.

2) Two major limitations were raised throughout the review process, namely i) the lack of mechanistic understanding of the generation of these large rhythmic depolarizations (LRDs) and ii) the possibility that LRDs are merely pathological activity found in cortical circuits affected by epileptic activity. However, I have the impression that both reviewers and authors agreed that these points are probably not easy to address experimentally due to the low probability of these events and the difficulty of obtaining tissue samples from human patients. These limitations should therefore be clearly mentioned and discussed in the article, which is now the case.

The overall relevance and significance of this study remains a matter of debate. Are LRDs primarily an epiphenomenon associated with pathological cortical circuits, or are they important network events that have a functional impact on human brain function? Future studies will be needed to answer these questions, but I think the results reported in this article would be of interest to the growing community of neuroscientists working on human cortical circuits at the cellular level.

One minor point: I'm not convinced that LRDs are truly rhythmic in their occurrence. In fact, it seems rather that LRDs show great variability in their inter-event intervals (see figure 1c). Consequently, I would like to suggest replacing the term "rhythmic" with "recurrent".

Reviewer #2 (Remarks to the Author):

No further comments.

Reply to Reviewers' comments

Response to Reviewer #1's comments:

Reply: We thank the reviewer for the support of publication.

REVIEWERS' COMMENTS:

Reviewer #1 (Remarks to the Author):

This article by Yang et al. is a revised version of an article I previously reviewed for Nature Communication. This new version is not very different from the previous one, but the authors have made some improvements and corrections that address the main remaining issues as much as possible. As a result, my assessment of the current version remains very similar to my initial assessment:

1) The experiments and data analysis are well conducted, and I have no concerns about the validity and quality of the data. Although not very innovative and classic in its design, this study presents solid and convincing results. In particular, I find the detailed comparison of the physiology and anatomy of LRD+ and LRD- interneurons remarkable, and overall, the results are novel and original.

2) Two major limitations were raised throughout the review process, namely i) the lack of mechanistic understanding of the generation of these large rhythmic depolarizations (LRDs) and ii) the possibility that LRDs are merely pathological activity found in cortical circuits affected by epileptic activity. However, I have the impression that both reviewers and authors agreed that these points are probably not easy to address experimentally due to the low probability of these events and the difficulty of obtaining tissue samples from human patients. These limitations should therefore be clearly mentioned and discussed in the article, which is now the case.

The overall relevance and significance of this study remains a matter of debate. Are LRDs primarily an epiphenomenon associated with pathological cortical circuits, or are they important network events that have a functional impact on human brain function? Future studies will be needed to

answer these questions, but I think the results reported in this article would be of interest to the growing community of neuroscientists working on human cortical circuits at the cellular level.

One minor point: I'm not convinced that LRDs are truly rhythmic in their occurrence. In fact, it seems rather that LRDs show great variability in their inter-event intervals (see figure 1c). Consequently, I would like to suggest replacing the term "rhythmic" with "recurrent".

Reply: We disagree with Reviewer #1's assessment that the events are not rhythmic. Our results demonstrate that these events occur within a physiological range of brain waves, displaying regular and rhythmic patterns. While the LRDs occurred at varying frequencies in different neurons, the small standard deviation of 0.18 Hz indicates that these frequencies remain within a narrow band. For instance, Fig. 1a illustrates LRDs occurring in a repeating, cyclical pattern, with consistent timing that confirms the rhythmic nature of the network activity.

Response to Reviewer #2's comments:

Reply: We thank the reviewer for the support of publication.

Reviewer #2 (Remarks to the Author):

No further comments.